# Non-catalytic hydrogenation of VO$_2$ in acid solution

Yuliang Chen[1], Zhaowu Wang[2,3], Shi Chen[1], Hui Ren[1], Liangxin Wang[1], Guobin Zhang[1], Yalin Lu[1], Jun Jiang [2], Chongwen Zou[1] & Yi Luo[2]

Hydrogenation is an effective way to tune the property of metal oxides. It can conventionally be performed by doping hydrogen into solid materials with noble-metal catalysis, high-temperature/pressure annealing treatment, or high-energy proton implantation in vacuum condition. Acid solution naturally provides a rich proton source, but it should cause corrosion rather than hydrogenation to metal oxides. Here we report a facile approach to hydrogenate monoclinic vanadium dioxide (VO$_2$) in acid solution at ambient condition by placing a small piece of low workfunction metal (Al, Cu, Ag, Zn, or Fe) on VO$_2$ surface. It is found that the attachment of a tiny metal particle (~1.0 mm) can lead to the complete hydrogenation of an entire wafer-size VO$_2$ (>2 inch). Moreover, with the right choice of the metal a two-step insulator–metal–insulator phase modulation can even be achieved. An electron–proton co-doping mechanism has been proposed and verified by the first-principles calculations.

[1] National Synchrotron Radiation Laboratory, University of Science and Technology of China, Hefei 230029, China. [2] Hefei National Laboratory for Physical Sciences at the Microscale, Collaborative Innovation Center of Chemistry for Energy Materials, CAS Center for Excellence in Nanoscience, School of Chemistry and Materials Science, University of Science and Technology of China, Hefei 230026, China. [3] School of Physics and Engineering, Henan University of Science and Technology, Henan Key Laboratory of Photoelectric Energy Storage Materials and Applications, Luoyang, 471023 Henan, China. Yuliang Chen and Zhaowu Wang contributed equally to this work. Correspondence and requests for materials should be addressed to J.J. (email: jiangj1@ustc.edu.cn) or to C.Z. (email: czou@ustc.edu.cn)

As a typical transition oxide, $VO_2$ has a pronounced metal–insulator transition (MIT) behavior at the critical temperature near 68 °C, accompanying by a sharp resistance change up to five orders of magnitude and marked infrared switching effect within sub-$ps$ time scale[1–5]. It has thus shown great potential for important applications in memory material[6,7], smart window[8,9] and ultra-fast optical switching device[10]. Many efforts have been devoted to improve the phase transition properties of metal oxides[11–16], including the hydrogenation treatment, which has been demonstrated to be an effective way to tune the property of metal oxides[17–21]. Recent experiments observed that H-incorporations in M-$VO_2$ could result in a very stable metallic phase at room temperature[19,22], giving excellent thermoelectric performance[23]. Moreover, further injecting H into the lightly doped M-$VO_2$ could create another insulating state at the heavily H-doping situation[20], enabling the control of MIT in a reversible and consecutive manner. Previous studies examined the thermodynamic and kinetic properties of H or Li doping in $VO_2$ lattice[22,24,25], showing that H atoms preferred to diffuse along the $c$-axis of rutile $VO_2$ or $a$-axis of monoclinic $VO_2$. Although the hydrogenation techniques available are not sustainable as they are conventionally performed with noble-metal (Au, Pt, Pd) catalysts, high-temperature/pressure annealing treatment or high-energy proton implantation in vacuum condition[22,26–28].

In this work, we report a facile approach to hydrogenate monoclinic $VO_2$ film in acid solution at ambient condition by placing a low workfunction metal particle (Al, Cu, Ag, Zn, or Fe) on $VO_2$ surface. The workfunction difference will cause electron flowing from metal particles to $VO_2$, which in turn drives surrounding solution protons to penetrate into $VO_2$ due to electrostatic attraction, resulting a stable H atoms doping. This process will not only stabilize the $VO_2$ lattice in acid, but also induce the modulation of phase transitions under ambience conditions, which should be of great potentials for material applications. An electron–proton co-doping mechanism has been proposed and this synergetic doping method will stimulate more simple and cost-effective atomic doping techniques in the future.

## Results

**Metal-acid treatment induced H-doped $VO_2$ film.** Is it possible to use acid solution as a natural proton source to achieve the hydrogenation of $VO_2$ at ambient condition? At the first glance, this appears to be an impossible mission, as the textbook tells us that pristine metallic oxides including $VO_2$ are easily dissolved in acid through the well-established reaction of $VO_2 + 4H^+ \rightarrow V^{4+} + 2H_2O$. Indeed, as shown in Fig. 1a, when a 30 nm M-$VO_2$/$Al_2O_3$ (0001) epitaxial film grown by molecular beam epitaxy method[29] (Supplementary Fig. 1) held by a plastic tweezers was put into a 2%wt $H_2SO_4$ acid solution, the yellowy $VO_2$ epitaxial film completely disappeared within 3 h. Although when a steel tweezers was used to hold the sample, as shown in Fig. 1b, the same $VO_2$ film suddenly demonstrated excellent anti-corrosion ability: it remains intact after 3 h in the acid solution. Scanning electron microscope (SEM) images in Fig. 1c show that the thickness and morphology of $VO_2$ film hardly change even after 20 h in acid solution. In addition, the atomic force microscope (AFM) measurements show nearly zero thickness variation for metal-acid-treated samples (Supplementary Fig. 2a), which is consistent with the SEM cross-section image and confirms the anti-corrosion ability. More convincingly, the trace element analysis in Fig. 1d revealed that the concentration of $V^{4+}$ cations in solution increased from 0.11 to 1.82 μg/ml after immersing a $VO_2$ film held by a plastic tweezers in acid from 30 min to 20 h, whereas it kept very low value at 0.03–0.06 μg/ml with a steel tweezers. All these results firmly point to the fact that the

attachment of a metal can give excellent anti-corrosion ability to $VO_2$.

Interestingly, the hydrogenated $VO_2$ produced by Au or Pd catalyst is found to be very stable in acid solution (Supplementary Fig. 2b). One can thus reasonably assume that the anti-corrosion ability of such metal-acid-treated $VO_2$ is due to the hydrogenation. The X-ray diffraction (XRD) spectra in Fig. 1e show the dynamic shifts of (020) diffraction peak from 39.8° to 36.7° after the metal-acid treatment due to the cell expansion caused by H-incorporation, which agree well with the results for lightly and heavily hydrogenated $VO_2$ through conventional noble-metal catalysis at high temperature (Supplementary Fig. 3a). These hydrogenated $VO_2$ films show successive metallic and insulator states as the hydrogen doping concentration increasing[30].

The X-ray photoelectron spectroscopy (XPS) measurements presented in Fig. 1f clearly indicate the conversion from $V^{4+}$ to $V^{3+}$ state as the result of H intercalation, which is further confirmed by the variations of O1$s$ peak at ~531.6 eV for the O–H species. The change of valence state from $V^{4+}$ to $V^{(4-\delta)+}$ or even to $V^{3+}$ state is also verified by the X-ray absorption near-edge structure (XANES) spectra in Fig. 1g as the V $L$-edge curves shift continuously to lower energy. After the metal-acid treatment, the relative intensity ratio of the $t_{2g}$ and $e_g$ peaks in O $K$-edge curves decreased substantially, reflecting the variation of electron occupancy due to electron doping as well as the loss of the $d_{//}$ state upon hydrogenation[31]. All these spectroscopic features induced by metal-acid treatment for 1.5 and 10 h agree well with corresponding measurements on lightly and heavily hydrogenated $VO_2$ through conventional catalysis techniques (Supplementary Fig. 3), respectively. It can thus be concluded that the metal-acid treatment can indeed create H-doping in the $VO_2$ film.

**Hydrogenation of a wafer-size $VO_2$ film.** It is noted that contact area between the metal tweezers and the $VO_2$ film is actually quite small. To further quantitatively explore the effect of metal attachment, we have placed a tiny Cu particle (~1.0 mm in diameter) at the center of one 2-inch M-$VO_2$/$Al_2O_3$(0001) epitaxial film, and immersed them together into 2%wt $H_2SO_4$ solution. It is observed in Fig. 2a that the bare $VO_2$ film with yellowy color could be dissolved within 1.5–3 h. In sharp contrast, the small copper particle has provided the protection for the whole 2-inch wafer from acid corrosion. In addition, when the Cu particle is taken away after the treatment, the film remains stable in acid solution as it has already been hydrogenated (Supplementary Fig. 4).

A key evidence about the hydrogenation of $VO_2$ is the insulator–metal transition at room temperature[19,22], i.e., the hydrogenation converts the insulated M-$VO_2$ to the metallic phase (Supplementary Fig. 5). In comparison with the original insulating M-$VO_2$ film (Fig. 2b), the surface resistance of above Cu-acid-treated sample is decreased by almost three orders of magnitude (Fig. 2c). After cyclically heating the sample in air between 40 and 90 °C for about 2 h, the intercalated hydrogens could be completely removed, and the film is recovered back to the insulated phase (Fig. 2d), which is consistent with the results obtained from hydrogenated samples through conventional catalysis (Supplementary Fig. 6). From Fig. 2a, one can note that the small copper particle can be taken away to leave out pure H-doping material. It is certainly a much better approach than the conventional catalysis-based technique as those metal catalysts (Au or Pt) sputtered onto the $VO_2$ film surface are hardly removable. In addition, the latter gives only limited hydrogenation area covered by catalysts (Supplementary Fig. 5b).

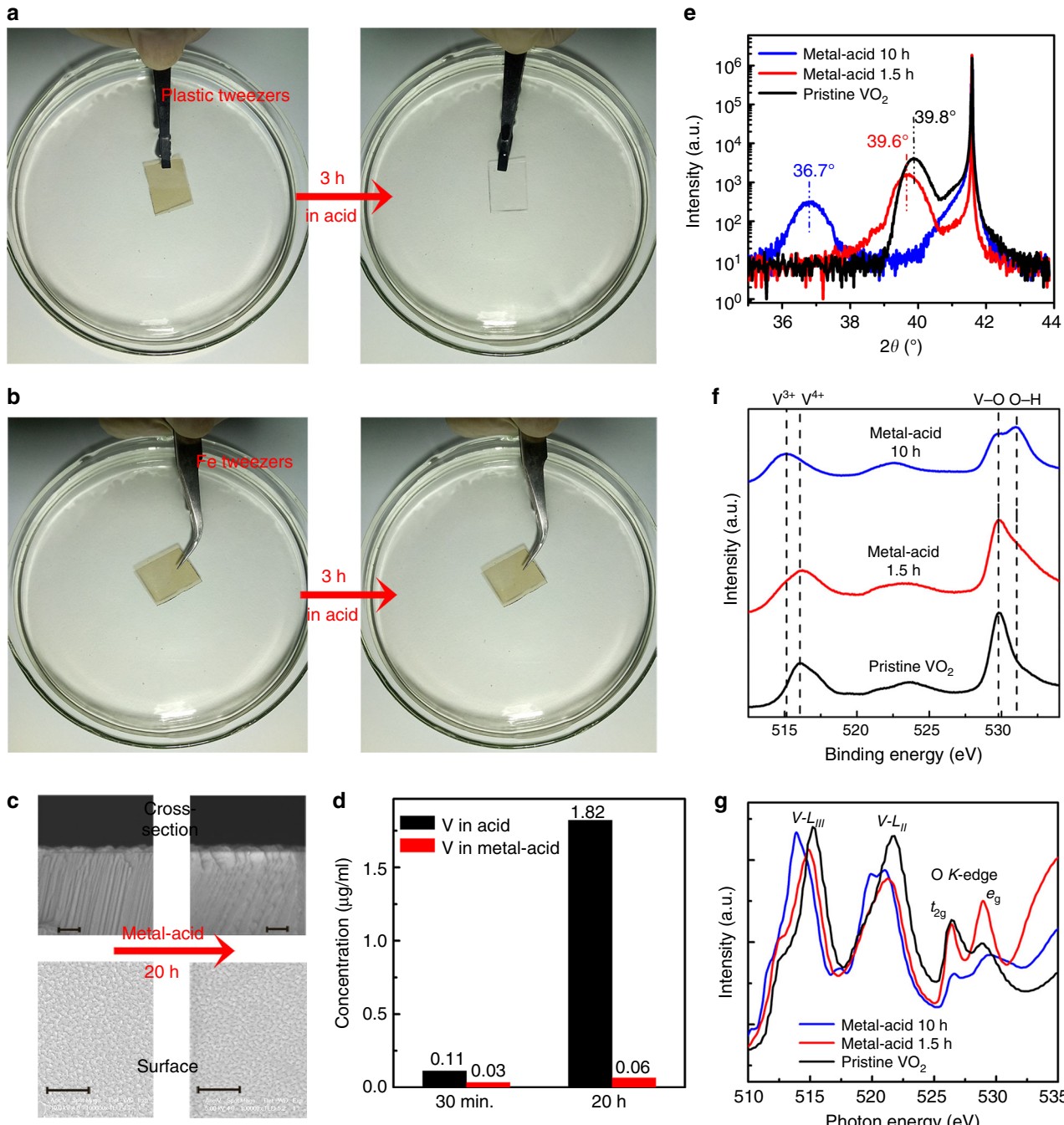

**Fig. 1** Metal-acid treatment induced hydrogenation of VO$_2$ film. **a** The VO$_2$ film on Al$_2$O$_3$ substrate held by a plastic tweezers was dissolved by 2%wt H$_2$SO$_4$ acid in 3 h. **b** Although a steel (Fe) tweezers attachment made the film intact in acid, showing pronounced anti-corrosion ability. **c** The SEM images for the cross-sections and surface morphologies of the VO$_2$ film being treated by metal (Fe)-acid for 20 h, showing that the VO$_2$ film maintains unchanged thickness and surface morphologies. The scale bar is 100 nm for the cross-sections and 500 nm for the surface morphologies, respectively. **d** Trace element analysis shows the V$^{4+}$ concentrations in solution changing from 0.11 to 1.82 μg/ml after 30 min to 20 h with acid treatment, whereas very low V$^{4+}$ concentration at 0.03–0.06 μg/ml is found at the same time period with metal (Fe)-acid treatment. The **e** XRD, **f** XPS, and **g** XANES characterizations for the pristine VO$_2$ and metal (Fe)-acid-treated samples for 1.5 and 10 h, respectively. The pronounced (020) XRD peak shift from 39.8° to 36.7°, the increased V$^{3+}$ and O–H XPS signals, and enhanced $e_g/t_{2g}$ XANES signal ratio (reflecting the variation of electron occupancy) along with the increase of metal (Fe)-acid time, indicate the lattice changes and O–H bonds formations due to light and heavy hydrogenations

**Hydrogenation effects controlled by different metals**. We have found that several other metals, such as Al, Cu, Ag, Zn, or Fe, can all induce hydrogenation and thereby protect VO$_2$ from corrosion in acid, whereas Au and Pt can not. The effects of different metals are illustrated in Fig. 3a. One important parameter associated with the choice of the metal is the workfunction. The

workfunction values calculated for the metals (Supplementary Fig. 7), VO$_2$ and H$_{0.5}$VO$_2$ are plotted together with the reported experimental values[32] in Fig. 3b. Simulations of pristine and hydrogenated VO$_2$ systems were based on the most stable atomic models obtained by previous studies[30]. Because of high lattice symmetry, the electronic structure of H-doped VO$_2$ is sensitive to

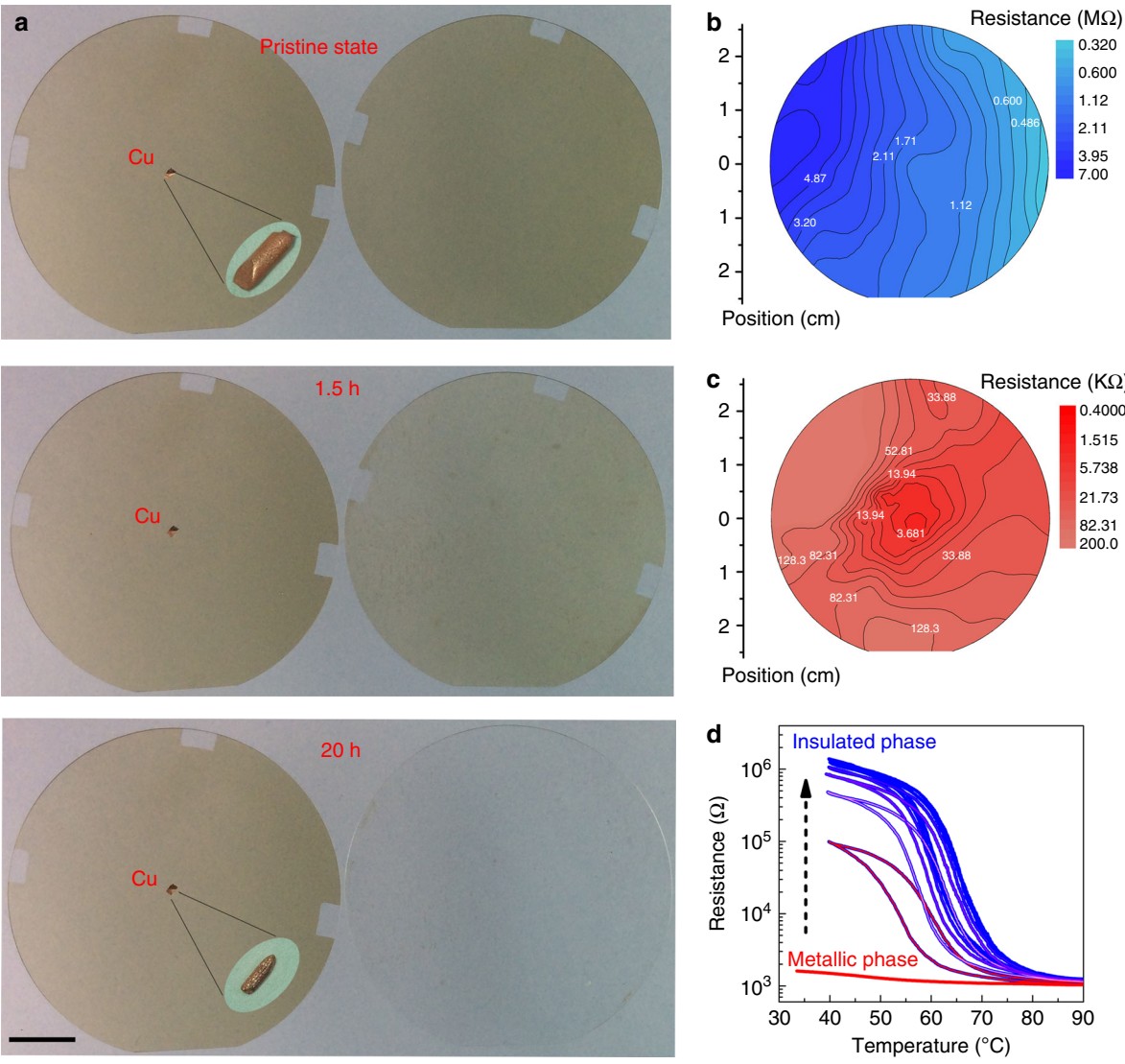

**Fig. 2** Hydrogenation of a wafer-size $VO_2$ film with a tiny Cu particle. **a** The two 2-inch $VO_2/Al_2O_3$ wafers immersed in 2%wt $H_2SO_4$ acid solution. The sample with a tiny copper (Cu) particle (~1 mm) attached on the center surface exhibits pronounced anti-corrosion ability, whereas the bare $VO_2/Al_2O_3$ film is completely corroded within 1.5–3 h, leaving the transparent $Al_2O_3$ substrate. The scale bar is 1 cm. **b** The resistance mapping for the 2-inch pristine $VO_2$ film. **c** The resistance mapping for the metal(Cu)-acid-treated $VO_2$. For the whole 2-inch wafer, the surface resistance is decreased by almost three orders of magnitude in comparison to the pristine film, reflecting the MIT of M-$VO_2$ by hydrogenation. **d** The resistance measurement in air for the metal (Cu)-acid-treated M-$VO_2$ as the function of heated temperature. Along with the pronounced hysteresis R–T curve, the metallic sample gradually recovers to the initial insulated M-phase $VO_2$

the H-doping concentration but not to the atomic sites of H in lattice. By testing all of the 16 possible H-doping sites (Supplementary Fig. 8; Supplementary Table 1), we have taken the one with lowest energy for further investigation. A clear pattern can be observed: with the respect to the workfunction of $VO_2$, the metal with smaller workfunction value can induce the hydrogenation. With such a workfunction difference, metals with higher electric Fermi level ($E_F$) would donate electrons to the interfaced $VO_2$ with lower $E_F$ (Fig. 3c). Calculations show that one ($1 \times 1$) $VO_2$ unit could extract $0.47$–$2.50 e^-$ from metals with lower workfunction (Fig. 3d; Supplementary Fig. 9; Supplementary Table 2). On the other hand, higher workfunction metals, Au and Pt, give nearly no extra electrons at the interface of M-$VO_2$. It should also be noted in Fig. 3b that Al and Zn metals hold even lower workfunction than the lightly H-doped system of $H_{0.5}VO_2$, suggesting the continuing donation of electrons from metal to lightly hydrogenated $VO_2$ which later attracts more hydrogen to

penetrate. Therefore, the final products of Al/Zn-acid treatment are heavily H-doped $VO_2$ with insulator phase while those of Ag/Cu-acid are conductive lightly H-doped $VO_2$, as validated by XRD, XPS, XANES, and Raman characterizations in Supplementary Fig. 3. These results thus demonstrate that the electron-rich $VO_2$ interface can attract and interact with the protons in acid solution, resulting in a feasible way to generate hydrogen atoms needed by hydrogenation.

By examining six $VO_2$ surface sites for a proton to adsorb (inset graph in Fig. 4a and Supplementary Fig. 10), it is found that more doped electrons lead to higher adsorption energies for all sites (Fig. 4a). For instance, on site 1, the proton adsorption energy of 3.68 eV in neutral circumstance is increased to 5.04 eV for a $VO_2$ unit with $4e^-$ charge. The doped electrons also promote the diffusion of surface hydrogens into the $VO_2$ crystal, with a possible migration pathway along the [100] direction (Supplementary Fig. 11). We can therefore propose an

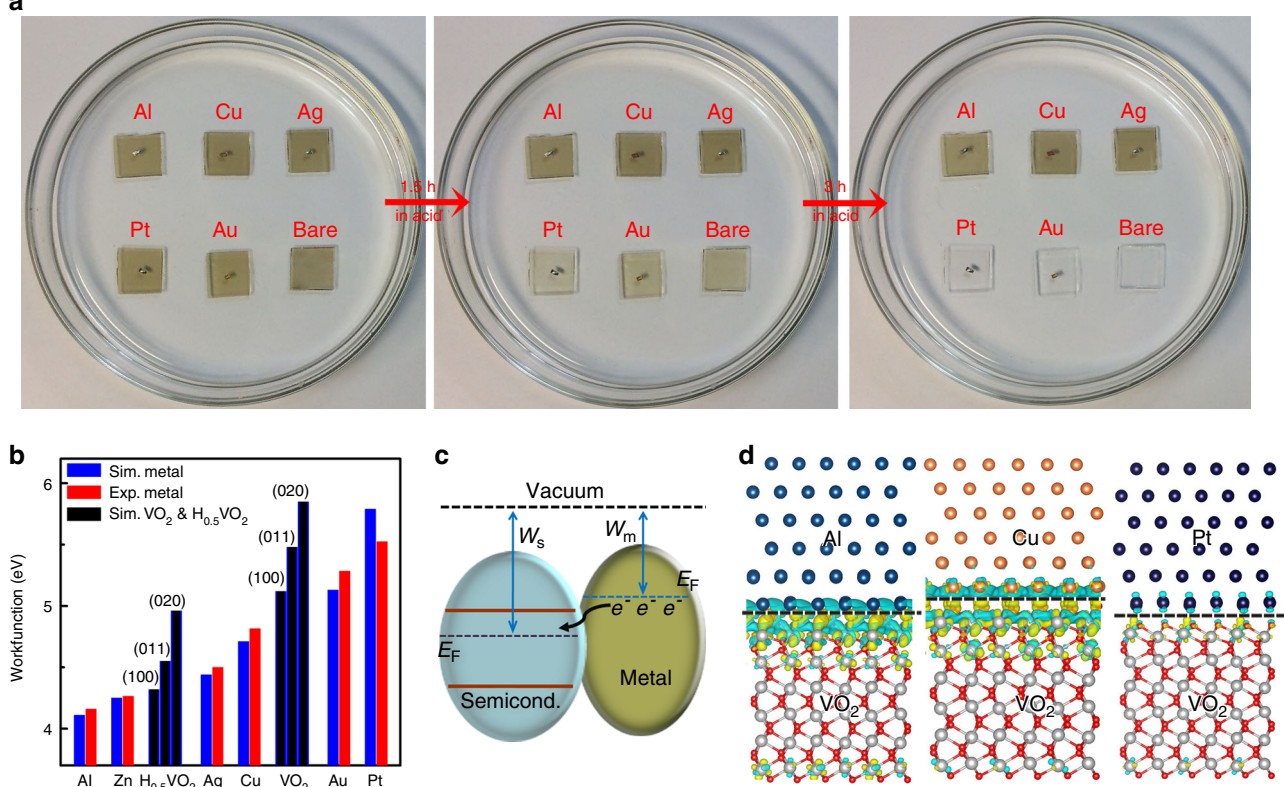

**Fig. 3** Hydrogenation effects induced by different metals. **a** Metals, such as Al, Cu, Ag, can protect M-VO$_2$ (1 cm × 1 cm size) from corrosion in 2%wt H$_2$SO$_4$ acid solution, whereas metals, Pt and Au, can not. **b** Computed and experimental workfunction ($W_F$) values for metals, VO$_2$, and lightly hydrogenated H$_{0.5}$VO$_2$, with the order of Pt > Au > VO$_2$ > Cu > Ag > H$_{0.5}$VO$_2$ > Zn > Al. **c** Schematic depiction of electrons flowing from metal with a higher Fermi level (i.e., lower workfunction $W_m$) to semiconductor with a lower Fermi level (i.e., higher workfunction $W_s$) at the interface. **d** Computed differential charge distribution at Al/Cu/Pt–VO$_2$(020) interfaces, showing that active metals (Al and Cu) donate effective electrons to VO$_2$. Green and yellow bubbles represent hole and electron charges, respectively. Gray, red, cyan, brown, navy beads stand for V, O, Al, Cu, Pt atoms, respectively

electron–proton co-doping strategy to create stable neutral H-doping in VO$_2$. More specifically, driving by the electrostatic attraction, the surrounding protons could penetrate into VO$_2$ to meet electrons, resulting in neutral H intercalation. The incorporation of H in the VO$_2$ crystal prohibits further attack/adsorption of protons to oxygen, and increases the formation energy required for oxygen vacancy defect (Supplementary Fig. 12), resulting in the anti-corrosion ability in acid solutions.

The H-doping changes the VO$_2$ electronic structures. For a VO$_2$ unit with small H-doping concentration of H$_{0.25}$VO$_2$ (Fig. 4b), the evolution of the electronic structure is reflected by the computed partial density of state (PDOS) of the V-3$d$ orbitals in Fig. 4b. The formation of H–O bonds causes electrons transferring from H to O atoms (Supplementary Table 3), which in turn promotes the electron occupancy of V-3$d$ orbitals. Such effects give rise to the up-shifting of Fermi level from the pure VO$_2$ to H$_{0.25}$VO$_2$ (Fig. 4b). Originally, VO$_2$ exhibits a typical insulating state, with wide energy gap consisting of fully occupied valence band and empty conduction band. The H-doping then makes the conduction band edge states partially occupied, as for the case of H$_{0.25}$VO$_2$.

The same concept can be also used to explain the contagious hydrogenation process that enables a ~1 mm metal particle to convert a 2-inch semiconductor wafer. As shown in Fig. 3b, the work functions of the lightly hydrogenated H$_{0.25}$VO$_2$ with three facets of (020), (011), (100) are 4.32–4.96 eV, which are all lower than those of pristine VO$_2$ around 5.12–5.85 eV. For any H-doped VO$_2$ parts created by metal-acid treatment, electrons

would flow/dope into neighboring unhoped VO$_2$ with lower Fermi level (Fig. 4c; Supplementary Fig. 13). The electron-rich area drives further proton penetration to the neighboring un-doped VO$_2$.

**Electron–proton co-doping mechanism.** A contagious spreading of electron–proton co-doping mechanism is summarized in Fig. 4d: firstly the metal with lower workfunction donates electrons to the interfaced VO$_2$ due to Fermi level difference, resulted in extra electrons accumulated in oxide layer; Then the doped electrons attract surrounding protons in acid solution to penetrate into the oxide semiconductor, creating H-doped structure at the top layer and causing the surface insulator-to-metal phase transition. Simultaneously, the attached metal particle is partially dissolved in acid, which balances the total charge in solution. This balance of charge is essential to drive the hydrogenation of VO$_2$ as this route:

$$M - x[e^-] \rightarrow M^{x+}, \tag{1}$$

$$VO_2 + x[e^-] + x[H^+] \rightarrow H_xVO_2. \tag{2}$$

Otherwise, the reaction will follow the route of VO$_2$ + 4H$^+$ → V$^{4+}$ + 2H$_2$O, resulting the corrosion of VO$_2$ in acid. The test of immersing only parts of a Cu/VO$_2$ sample into acid without metal in solution causes no anticorrosive property (Supplementary Fig. 14), clearly showing this situation. After the above stage,

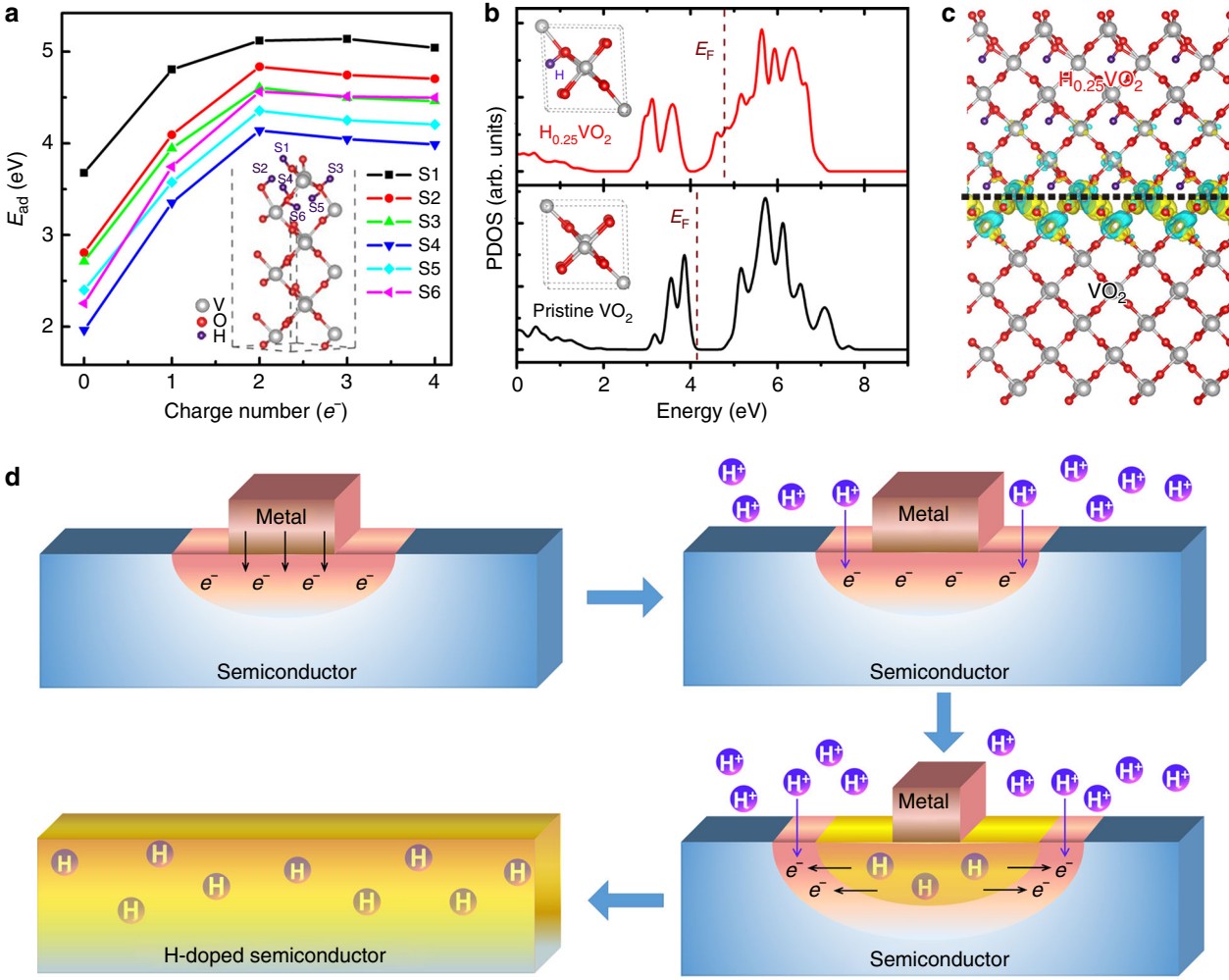

**Fig. 4** Electron–proton co-doping mechanism. **a** Computed adsorption energies for a proton to six adsorption sites of $VO_2$ (020) surface, increased with the increasing amount of doped electrons. **b** Evolutions of V-3$d$ partial density of state (PDOS), suggest the change of semiconductor band gap in the insulated pristine $VO_2$ to the zero energy gap in $H_{0.25}VO_2$. Fermi level is marked with purple dashed lines. **c** Computed differential charge distribution at $H_{0.25}VO_2$–$VO_2$ interface, showing each $H_{0.25}VO_2$ supercell donated ~2.06$e^-$ to un-doped $VO_2$. Here green and yellow bubbles represent hole and electron charges, respectively. **d** The schematic illustration of the contagious electron–proton co-doping mechanism with the metal-acid treatment to semiconductor: firstly the electrons flow to semiconductor when the metal contacts $VO_2$ film. Once the metal/$VO_2$ is immersing into acid solution, chemical reactions go sequentially as $M - x[e^-] \rightarrow M^{x+}$ and $VO_2 + x[e^-] + x[H^+] \rightarrow H_xVO_2$. Here protons penetrate to meet electrons, creating conductive H-doped structure. Meanwhile the attached metal (M) is gradually dissolved in acid to become $M^+$ cations for balancing charges in solution. Then the electrons flow from conductive H-doped structure to the un-doped parts, driving more proton penetration. Finally the repeated electron flowing–proton penetration–phase transition–electron further flowing cycle expands toward full H-doping

the conductive H-doped structures delivery electrons to adjacent un-doped $VO_2$ parts, triggering the next round of electron–proton co-doping and insulator-to-metal phase transition. Finally, the repeated electron flowing–proton penetration–phase transition–electron further flowing cycle expands toward full H-doped oxide material. This spreading of co-doping from metal center to edge is reflected by the onion-like contour map of resistance after metal-acid treatment (Fig. 2c), which decreases gradually from the metal-attached center to the wafer edge. In addition, the hydrogenation process is also gradually completed from the top to bottom layer with considering the time-dependent hydrogenation-related Raman or XRD signals (Supplementary Fig. 15).

It should be pointed out that since the corrosion of $VO_2$ caused by oxygen atom moving out of lattice is much slower than the migrations of electrons or protons, the dynamics of this co-doping mechanism ensure the quick hydrogenation of $VO_2$ surface before being corroded by acid, resulting in the anti-

corrosion property of wafer-size $VO_2$ film even at the beginning stage since we found the distinct resistance decrease of $VO_2$ with several seconds metal-acid treatment.

## Discussion

On the basis of the proposed concept, one can anticipate that the metals with very low workfunction, such as Al or Zn, can lead to heavy hydrogenation of $VO_2$ films in acid solution. It means that the induced metallic state would eventually be transferred into another new insulating state because of nearly saturated hydrogenation (Supplementary Fig. 3; Supplementary Fig. 16), which agree well with the different H concentrations revealed by secondary-ion mass spectrometry measurement (Supplementary Fig. 17). This observation is consistent with very recent findings of the consecutive insulator–metal–insulator transitions induced by increasing H-doping concentration[20]. In addition, based on this metal-acid treatment, partially hydrogenation process with a

selected region can also be easily achieved by control the immersing area in acid solution (Supplementary Fig. 18). Remarkably, this simple metal-acid treatment is found to be a universal strategy that can be extended to doping ions in general. Replacing the acid solution by polymeric solution with $Li^+$, metallic Li-doped $VO_2$ films can also be obtained (Supplementary Fig. 19).

The ability to hydrogenate $VO_2$ with protons in acid solution demonstrated here provides a facile strategy to induce phase modulation of $VO_2$ materials, and the later successful doping of $Li^+$ into $VO_2$ suggests a general atomic doping approach of using proton or cation solvent sources together with electrons from metals. It is a sustainable approach that operates at ambient condition in an environment friendly manner by completely avoiding the use of precious catalysts and high-energy consumptions. The doping concept established in this study will have strong impact on the development of new functional materials in different applications.

## Methods

**Thin-film growth**. The 2-inch wafer-size $VO_2$ (020) epitaxial films were grown on c-cut sapphire by an rf-plasma assisted oxide molecular beam epitaxy (rf-OMBE) equipment and more details for the film preparation are reported elsewhere[29].

**Conventional hydrogenation treatment for $VO_2$ film**. Nano-sized Au islands were sputtered on M-$VO_2$ surface as the catalysis. The Au/$VO_2$ samples were annealed in tube furnace with the forming gas (15% $H_2$/85% Ar) under various conditions. The lightly doped metallic H-$VO_2$ (120 °C for 2 h) and heavily doped insulated H-$VO_2$ (180 °C for 10 h) were prepared.

**Characterizations**. The resistance as the function of temperature was examined by Keithley 2400 sourcemeter with a variable temperature stage. For all of the measurements, the temperature sweeping rate was set at 0.1 K/s; The resistance distribution mapping for the prepared wafer-size $VO_2$ film were tested on room temperature and 120 °C, respectively. The cross-section and surface morphologies were investigated by Scanning Electron Microscopy (Gemini Fe-SEM 500 and FEI Sirion 200). Raman spectra were recorded by an integrated laser Raman system (LABRAM HR, Jobin Yvon). The 632.8 nm He–Ne laser was used as the excitation source. To obtain direct information about the hydrogen concentration of the metal-acid $VO_2$ film sample, the secondary-ion mass spectrometry (SIMS) measurements (Quad PHI6600) were conducted. The ICP-AES equipment (Optima 7300DV) was used to trace element concentration. The wavelength of Cu, V adopted 327.393, 290.880 nm respectively. The emission power is 1250 w. The distinguishability reached to 0.003 nm at 200 nm, and the detection limit is 4 ng/ml.

**Synchrotron-based measurements**. Synchrotron X-ray diffraction spectra, including $\theta - 2\theta$, X-ray reflectivity (XRR), $\Phi$-scan, rocking-curve, were conducted at the BL14B beamline of the Shanghai Synchrotron Radiation Facility (SSRF). The SSRF is a third-generate accelerator with a 3.5 GeV storage ring. The BL14B beamline shows the energy resolution ($\Delta E/E$) of $1.5 \times 10^{-4}$ @10 keV and the beam size of $0.3 \times 0.35$ mm with the photo flux of up to $2 \times 10^{12}$ phs/s@10 keV. Considering the photo flux distribution and the resolution, the 0.12398 nm X-ray was chosen during the experiment.

The X-ray absorption near-edge spectroscopy (XANES) was conducted at the XMCD beamline (BL12B) in National synchrotron radiation laboratory (NSRL), Hefei. The total electron yield (TEY) mode was applied to collect the sample drain current under a vacuum better than $3.75 \times 10^{-10}$ Torr. The energy range is 100–1000 eV with an energy resolution of ca. 0.2 eV. The X-ray incident angle is 54.7°. During the measurement, the samples were firmly adhered on the conductive substrate with random orientation, so the polarization dependence is not considered.

The X-ray photoelectron spectroscopy (XPS) beamline was conducted in National synchrotron radiation laboratory (NSRL), Hefei. The photoemission beamline covers photon energies from 100 to 1000 eV with a typical photon flux of $1 \times 10^{10}$ phs/s and a resolution ($E/\Delta E$) better than 1000 at 244 eV. The analysis chamber is connected to the beamline and equipped with a VG Scienta R3000 electron energy analyzer, and the base pressure is $1.5 \times 10^{-10}$ Torr.

**First-principles calculations**. All calculations are performed with density functional theory (DFT), using the Vienna ab initio simulation package (VASP) code[33]. The exchange and correlation terms are described using general gradient approximation (GGA) in the scheme of Perdew–Burke–Ernzerhof (PBE)[34]. Core electrons are described by pseudopotentials generated from the projector

augmented-wave method[35], and valence electrons are expanded in a plane-wave basis set with an energy cutoff of 480 eV. Slab model method is used to model the $VO_2$ surface and metal–$VO_2$ interface, the thickness of vacuum are larger than 15 Å. The DFT+U method is employed to optimize the structure, U and J are chosen to be 4 and 0.68 eV. The geometry relaxation is carried out until all forces on the free ions are converged to 0.01 eV/Å. In the calculation of electronic structures, DFT with hybrid functionals proposed by Heyd, Scuseria, and Ernzerhof (HSE06) is used[36]. Climbing image nudged elastic band (CI-NEB) method[37] is used to find the minimum energy paths and the transition states for diffusion of H from surface to subsurface, with a force converge <0.05 eV/Å.

**Data availability**. The remaining data contained within the paper and supplementary files are available from the authors upon request.

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

## Acknowledgements

This work was partially supported by the National Basic Research Program of China (2014CB848900), the National Key Research and Development Program of China (2016YFA0401004), the National Natural Science Foundation of China (U1432249, 11574279, 11404095, 21633006, 11704362), the Fundamental Research Funds for the Central Universities; the funding supported by the Youth Innovation Promotion Association CAS and the Open Research Fund of State Key Laboratory of Pulsed Power Laser Technology, Electronic Engineering Institute. We also acknowledge supports from the X-ray diffraction beamline (BL14B1) in Shanghai Synchrotron Radiation Facility, the XMCD beamline (BL12B) and photoelectron spectroscopy beamline (BL10B) in National Synchrotron Radiation Laboratory (NSRL) of Hefei.

## Author contributions

Y.C., J.J. and C.Z. conceived the study. Y.C. and C.Z. designed the experiment and performed the initial tests. Z.W. and J.J. conducted the theoretical calculations. Y.C., S.C., H.R., L.W., G.Z. and Y.L. conducted the synchrotron-based measurements. Y.C., J.J., C.Z. and Y.L. wrote the manuscript. All authors discussed the results and commented on the manuscript.

## Additional information

**Competing interests:** The authors declare no competing interests.

