## [Peer Review File · Nature Communications]

Reviewers' comments:

Reviewer #1 (Remarks to the Author):

The manuscript under consideration presents intriguing studies on hydrogenation of VO₂ facilitated by interfacing an epitaxially growth thin film with low work function metal particles. Notably, the fundamental concept illustrated here was first demonstrated by Son et al. in Nature Materials 15, 1113–1119 (2016) and Natelson and co-workers in J. Am. Chem. Soc., 2014, 136 (22), pp 8100–8109. Some of the earliest works dates back to reports by Moskovits using VO₂/Pt as a hydrogen sensor. The manuscript seeks to elaborate upon current mechanistic understanding of the hydrogen intercalation phenomena and to establish its generalizability to different metals. Interfacing with low work function metals brings about a pronounced modulation of the conductance, which is ascribed to hydrogenation-induced metallization. The metallization is proposed to result from an alternative reaction pathway for electron-rich VO₂ as compared to bare VO₂, which is instead corroded under acidic conditions.

The synthetic methods will be of some utility even as it remains debatable whether the manuscript presents sufficient novelty in light of extensive prior work on metal-particle-mediated hydrogenation (ascribed to spillover mechanisms). In addition, I have enumerated several concerns below:

- (1) The characterization of the prepared hydrogenated VO₂ is rather incomplete and inadequate. The authors seem to simply want to reference their data to “lightly hydrogenated” and “heavily hydrogenated” prepared by noble metal catalysis instead of providing an unequivocal discussion of structure and site occupancies. For instance, is the hydrogenated material orthorhombic or rutile VO₂ or some other phase? A clear structure elucidation needs to be performed using diffraction or Raman spectroscopy methods.
- (2) Why is a theta-2-theta scan omitted for the material after hydrogenation?
- (3) What are the specific H sites in VO₂? Has there been a rigorous examination of different possible intercalation sites? What are the relative energetics of the sites examined?
- (4) None of the characterization methods directly address hydrogen sites-NMR methods or spectroscopic methods would be of some utility here.

(5) What are the migration pathways through the monoclinic lattice (work by Mazhos in Phys. Chem. Chem. Phys., 2017,19, 22538-22545 may provide some useful ideas on possible diffusion pathways).

(6) Can the authors provide a balanced equation for the hydrogenation reaction? I am having trouble seeing how charges and protons balance across the entire construct and the simplistic schematic in Figure 4 does not provide a clear answer.

(7) Are the interstitial species H^+ or H radicals reduced one electron?

Since the work is not really the first report of hydrogenation of VO_2 , a more detailed mechanistic perspective rather than ascribing the observed behavior to a work function difference ought to be necessary.

Reviewer #2 (Remarks to the Author):

The manuscript, entitled to “Non-catalytic hydrogenation of VO_2 in acid solution” by Chen et al., reported simple approach for hydrogenation of VO_2 without commonly used catalyst. Utilizing acid solution as a rich source of protons, complete hydrogenation of wafer-size VO_2 could be achieved by placing a small piece of appropriate metal on the VO_2 surface without corrosion of VO_2 by acid solution. To explain this interesting phenomena, the authors proposed an electron-proton co-doping mechanism, which is verified by the first principles calculations.

Overall, this article shows unique non-catalytic hydrogenation approach by controlling work function and electron transfer to VO_2 thin films. The result on non-catalytic hydrogenation using acid solution have not been reported before, although non-catalytic approach using atomic hydrogen gas (Nano Lett., 14, 5445 (2014)) or ionic liquid by gating. The role of electron transfer was emphasized by utilizing metal with low work function on VO_2 films, which can be further applicable to the development of new approach for band-filling control of strongly correlated VO_2 films. Therefore, this report will attract the attention of other researcher in materials science community and will potentially extend the scope of physics into the efficient band-filling control of correlated phase using hydrogen dopants. But, I have several technical comments on this manuscript, which should be addressed before the publication in Nature Communication.

1) The authors argued that H intercalation occurred in the VO₂ lattice after metal-acid treatment by showing the “indirect” methods (e.g., lattice expansion by XRD, resistance change by R-T measurement, valence change by XPS and XANES). But, those methods do not directly detect hydrogen atoms. Therefore, it is necessary to “directly” quantify the amount of hydrogen incorporated in thin films after hydrogenation. I would suggest using previously reported techniques, such as dynamic SIMS or RBS (See Nature 546, 124 (2017), Nature Mater. 15, 1113 (016)), as direct evidence of H intercalation in metal-acid treated VO₂ film.

2) The authors reported that all experiment using metal-attached VO₂ film, which showed anti-corrosion property, were performed in 2% wt H₂SO₄ acid with same pH condition. I am wondering if the authors tried doing same experiment by changing pH of H₂SO₄ acid.

3) The authors argued that, after annealing Li-VO₂ film, the Li atoms was driven out from the VO₂ lattice and congregated on the top-most surface of the film as shown in Fig. S15. I suspected the formation of Li oxide on the surface, instead of congregation of Li atoms on the surfaces. How could the author rule out this effect on the surface?

4) Fig. 2c showed that whole 2-inch VO₂/Al₂O₃ wafers become metallic uniformly by Cu-acid treatment. I am wondering if wafer-scale uniform insulator-metal transition also occurs in other metal-acid with Al or Zn. Furthermore, do you have any data on the wafer-scale resistance mapping of fully hydrogenated insulating VO₂/Al₂O₃ wafers formed by metal-acid treatment, which can be only shown in the manuscript on resistance mapping of pristine VO₂ films (Fig. 2b) and metallic hydrogenated VO₂ (Fig. 2c)?

5) Are all the samples in Fig. 1e, f, g, which is characterized by XRD, XPS, XANES, hydrogenated using steel (Fe) tweezers? Figure caption is unclear. If this was hydrogenated using Fe tweezers, was samples fully hydrogenated? In terms of XRD, XPS, XANES, samples seem to become fully hydrogenated insulating phase, but the work function of Fe is similar to Cu and is higher than Al/Zn. Therefore, I should expect less effective hydrogenation process of Fe-acid treatment than Al/Zn-acid treatment, but the effectiveness of Fe-acid hydrogenation appears to be similar to that of Al/Zn-acid hydrogenation (Please compare Fig 1e, f, g and Fig. S3).

6) Small typo: the blue line in Fig. 1e, which was shifted more than the red line from the peak of pristine sample, should indicate metal-acid treated samples for 10 hours, not metal-acid treated samples for 1.5 hours.

Reviewer #3 (Remarks to the Author):

Chen et al. presented a new way of hydrogenating VO₂ by using acid solution. The method is novel and unique. By taking advantage of low work-function metal, large-scale hydrogenation of VO₂ can be achieved. Overall, the experimental result is interesting. However, there are a few concerns that need to be addressed before publication.

1. The paper emphasizes on the hydrogenation of VO₂ film, but also mentioned that using acid might etch the surface. This argument is supported using concentration data (Figure 1d) of dissolved V ions. It will be more intuitive and straightforward to show the thickness change as a function of immersion time of VO₂ dipping in acid by using atomic force microscope.
2. The authors claim that it may also work for other oxides. This assertion should be deleted unless they can demonstrate the same phenomenon in different oxides, such as ZnO, or TiO₂, etc. Comparing to other oxides VO₂ is unique because the strong correlation of electrons makes it a model system of Mott insulator. Many properties are rooted from this aspect.
3. In line 78, the authors claim that the anticorrosion of VO₂ is due to the hydrogenation. So the question is that at the very beginning of the reaction, the metal is just in contact with the film and the hydrogenation hasn't started yet, is it still anticorrosive? What would be the dynamics of hydrogenation caused anticorrosion or how the anticorrosion ability evolves during the hydrogenation process?
4. In Figure 1e, the blue color line should correspond to 10 hours treatment, not the 1.5 treatment.
5. In line 99, the authors immersed metal together with the film, which means the metal also contacts with the acid directly. Here brings the question: will there be any possibility that metal dissolves into the acid and releases metal ions, which stops the corrosion of VO₂? The authors need to rule out this possibility by performing experiment that half of the film contacted with metal left outside of the liquid and the other half from the same piece of film immerses into the acid to see if the hydrogenation can still be observed.

6. English need to be polished. For example, the sentence in line 114 is confusing. Single and uniform hydrogenated VO₂ film can be achieved by removing copper metal after metal-acid treatment. Line 59, “nature” needs to be changed to “natural”. Line 67, “magically” needs to be deleted to fit into a writing style of scientific research paper.

7. In line 170, the mechanism needs to be thought carefully. The authors propose a contagious spreading mechanism of hydrogenation. Does that mean the hydrogenation is spreading from the center of metal? Is there any proof of that? In contrast to this theory, a global hydrogenation process may happen at the film surface, because electrons transferred from metal to VO₂ surface and should expend to the whole surface instantaneously, not like protons which have much less mobility. Although monoclinic VO₂ is regarded as an insulator, it is still quite conductive comparing to conventional insulators, therefore, electrostatically charge can move easily on M-VO₂. Therefore, more experimental evidence are needed to approve the mechanism authors proposed.

8. In line 178 and 179, the authors conducted an experiment by continuously ramping the temperature during the hydrogenation, including the temperature above the phase transition temperature (67 degree). It complicates the experiment by mixing acid-hydrogenation and intrinsic phase transition because above 67 degree, the metallic phase of VO₂ will certainly cause dramatic hydrogenation due to the more opening rutile lattice structure as discovered by other researchers. So the authors should conduct another experiment by keeping the temperature at room temperature during the acid-hydrogenation, and take the sample out of acid for inspection of metal insulating transition by ramping the temperature. Repeated this process multiple times at each stage of hydrogenation.

Responses to Reviewers:

=====

Reviewer #1 (Remarks to the Author):

The manuscript under consideration **presents intriguing studies** on hydrogenation of VO₂ facilitated by interfacing an epitaxially growth thin film with low work function metal particles. Notably, the fundamental concept illustrated here was first demonstrated by Son et al. in Nature Materials 15, 1113–1119 (2016) and Natelson and co-workers in J. Am. Chem. Soc., 2014, 136 (22), pp 8100–8109. Some of the earliest works dates back to reports by Moskovits using VO₂/Pt as a hydrogen sensor. The manuscript seeks to elaborate upon current mechanistic understanding of the hydrogen intercalation phenomena and to establish its generalizability to different metals. Interfacing with low work function metals brings about a pronounced modulation of the conductance, which is ascribed to hydrogenation-induced metallization. The metallization is proposed to result from an alternative reaction pathway for electron-rich VO₂ as compared to bare VO₂, which is instead corroded under acidic conditions.

The synthetic methods will be of some utility even as it remains debatable whether the manuscript presents sufficient novelty in light of extensive prior work on metal-particle-mediated hydrogenation (ascribed to spillover mechanisms).

---We are very grateful to the reviewer for the encouraging comments. We also agree with the reviewer that studies of hydrogenating VO₂ and the related phase transition modulations have been performed frequently. While in all previous studies, the hydrogenating treatments were done with harsh conditions such as noble metal catalysis (normally Au, Pt or Pd), high-temperature/pressure annealing or high-energy H-implantation in vacuum.

In this study, we were the first to achieve hydrogenation of VO₂ film by using the acid solution as the proton source under a very facile condition. This also brings out a novel mechanism of electron-hydrogen co-doping, which leads to a non-catalytic hydrogenation approach. Thanks for the reviewer's comment, we now understand more clearly about the novelty of this work, and have now added these statements in the revised manuscript. Please refer to the **last paragraph of the main text**: *“The ability to hydrogenate VO₂ with protons in acid solution demonstrated here provides a new strategy to induce phase modulation of VO₂ materials, and the later successful doping of Li⁺ into VO₂ suggests a new atomic doping approach of using proton or cation solvent sources together with electrons from metals. It is a sustainable approach that operates at ambient condition in an environment-friendly manner by completely avoiding the use of precious catalysts and high energy consumptions. The groundbreaking new concept established in this study will have strong impact on the development of new functional materials in different applications.”*

In addition, I have enumerated several concerns below:

(1) The characterization of the prepared hydrogenated VO₂ is rather incomplete and inadequate. The authors seem to simply want to reference their data to “lightly hydrogenated” and “heavily hydrogenated” prepared by noble metal catalysis instead of providing an unequivocal discussion of structure and site occupancies. For instead, is the hydrogenated material orthorhombic or rutile VO₂ or some other phase? A clear structure elucidation needs to be performed using diffraction or Raman spectroscopy methods.

---- We thank the reviewer for pointing out this issue, which helps greatly to improve our presentations. We totally agree that systematical characterizations for the prepared hydrogenated VO₂ is important to determine the role of hydrogenation treatment.

Thus besides the diffraction measurements shown in Fig.1e & Fig. S3, we have further conducted the Raman tests for both the metal-acid treated VO₂ films and the hydrogenated VO₂ prepared by noble metal catalysis as shown in the following:

Figures: Raman characterizations for the metal-acid treated VO₂ (Left) and for the hydrogenated VO₂ prepared by noble metal catalysis (Right). For comparison, the Raman spectra for the rutile VO₂ film at 95°C was also recorded.

From the Raman measurements, it was observed that the Raman peaks at 192 cm⁻¹, 223 cm⁻¹ and 617 cm⁻¹ in pristine M-VO₂ were completely disappeared for the hydrogenated VO₂, similar as the Raman test from Rutile VO₂ phase. From both the diffraction and Raman tests, it was suggested that the hydrogenated VO₂ film shows a rutile-like phase structure and the lattice parameters are sensitive to the H-doping concentration. This is consistent with our previous work (Phys. Rev. B 96, 125130 (2017)) and other reports [Nature Materials 15, 1113–1119 (2016); Nat. Nanotechnol. 7, 357 (2012); Nano Lett. 14, 5445(2014)].

Please refer to **the red texts in Line 1-6/Paragraph 1/Page 6** of the revised manuscript: “It should also be noted in Fig. 3b that Al and Zn metals hold even lower

workfunction than the lightly H-doped system of $H_{0.5}VO_2$, suggesting the continuing donation of electrons from metal to lightly hydrogenated VO_2 which later attracts more hydrogen to penetrate. Therefore, the final products of Al/Zn-acid treatment are heavily H-doped VO_2 with insulator phase while those of Ag/Cu-acid are conductive lightly H-doped VO_2 , as validated by XRD, XPS, XANES and Raman characterizations in Fig. S3.”

Actually, in our previous work of hydrogenating VO_2 films by conventional noble metal catalysis (Phys. Rev. B 96, 125130 (2017)), we were able to measure the in-situ XRD map along with the change from lightly to heavily hydrogenating processes. It is revealed that under the noble metal catalysis, VO_2 film can be continuously hydrogenated in H_2 . As it is not affordable to do in-situ XRD measurement during metal-acid treatment, we have now cited our recent work of characterizing structures and properties of VO_2 with low and high hydrogen concentration (Ref. 29: Phys. Rev. B 96, 125130 (2017)). In the “Supporting Information” file, we also added the XRD, XPS, XANES and Raman results into Fig.S3 and added some sentences for the detailed descriptions.

Figures: The in-situ XRD tests for H-doped VO_2 film with conventional noble metal catalysis

(2) Why is a theta-2-theta scan omitted for the material after hydrogenation?

---- Actually, the theta-2-theta scan curves for the VO_2 film sample after hydrogenating treatment for 1.5 and 10 hours have already been displayed in **Figure 1e**. Here we should point out that after the metal-acid treatment by 1.5 or 10 hours, the VO_2 film was hydrogenated to different extents.

(3) What are the specific H sites in VO_2 ? Has there been a rigorous examination of

different possible intercalation sites? What are the relative energetics of the sites examined?

----We thank the reviewer for bringing out this issue. It is known that H atoms can be easily intercalated into VO₂ lattices to form –OH species, as confirmed by our XPS measurements in **Figure 1f**. Actually, we have just done first-principle simulations to test all possible H-doping sites and H_xVO₂ configurations in a previous work [Phys. Rev.B 96, 125130 (2017)], with results showing below. It is found that, the electronic structure of H-doped VO₂ is sensitive to the H-doping concentration but not the atomic sites of H in VO₂, main due to the lattice symmetry of the rutile-like structure for the H-doped VO₂.

Figures: (a~c)The possible atomic configurations of H-doped VO₂; (d~f) The DOS variation upon the H-doping; (g~i) the related band structure evolution after hydrogenation. [From PRB 96, 125130 (2017)]

We have now systematically examined the geometry models and density of state (DOS) of the HV₄O₈ system with one H at all of the 16 possible interstitial sites of a V₄O₈ unit cell, as shown below. Based on the computed total energy (eV) of a V₄O₈ unit cell with H-doping at 16 different sites (**Supporting information Table S3**), we extracted the one with the lowest energy for further electronic structure and property simulations. It is obvious that their electronic structures do not vary significantly and both of them show metallic properties. Therefore, we are rather confident in using the electronic structure of the most stable structure.

Figures: Computed geometry models and density of state (DOS) of the HV_4O_8 system with one H at different interstitial sites.

Nevertheless, in this specific work, we are mainly aiming at a new way of using protons in acid to be doped into VO_2 film through facile metal-acid treatment. The H atoms in VO_2 lattices are reported by many literatures [Nature Materials 15, 1113–1119 (2016); PRB 96, 125130 (2017); Nat. Nanotechnol. 7, 357 (2012)]. Therefore, since the detailed geometry models and density of state (DOS) of the H-doped system with H at different interstitial sites have already been reported in previous works, we have now added descriptions in **the red texts in Line 6-10/Paragraph 2/Page 5** of the revised manuscript: “*Simulations of pristine and hydrogenated VO_2 systems were based on the most stable atomic models obtained by previous studies. Because of high lattice symmetry, the electronic structure of H-doped VO_2 is sensitive to the H-doping concentration but not to the atomic sites of H in lattice. By testing all of the 16 possible H-doping sites (Table S3), we have taken the one with lowest energy for further investigation.*”

(4) None of the characterization methods directly address hydrogen sites-NMR methods or spectroscopic methods would be of some utility here.

---We thank the reviewer for pointing out this problem. We admit that NMR methods would be useful to detect the hydrogen sites, especially for liquid or organic sample. Unfortunately, it is currently too difficult to conduct NRM measurement on our VO_2 samples: Firstly, there is no definite chemical formula for the H-doped VO_2 material, as the metallic phase is related to the H_xVO_2 ($0 < x < 1$) with different doping concentration, while further hydrogenation treatment up to the saturation point of

H_1VO_2 leads to a new insulating state at room temperature; Secondly, our work are based on ultra-thin VO_2 film (tens of nanometers thickness), making it extremely hard to collect clear NMR data for extracting exact hydrogen site information.

On the other hand, as in **Line 8-10/Paragraph 2/Page 5 in the main text:** “Because of high lattice symmetry, the electronic structure of H-doped VO_2 is sensitive to the H-doping concentration but not to the atomic sites of H in lattice (Ref. 29: PRB 96, 125130 (2017)).” Therefore, the main point of this work is to apply characterization tools including the XPS, XANES XRD, and Raman spectra as well as the theoretical calculations, to supply evidences of H intercalations in lattice. These are also major characterizations conducted in previous experiments [Nature Materials 15, 1113–1119 (2016); J. Am. Chem. Soc. 136, 8100 (2014); Nat. Nanotechnol. 7, 357 (2012)].

Moreover, we have recently performed SIMS (secondary-ion mass spectrometry) characterization to directly examine the existence of H in VO_2 film, as shown in the following image. Please refer to the new **Supplementary figure S15 and Line 3-6/Paragraph 2/Page 8 in the main text:** “It means that the induced metallic state would eventually be transferred into another new insulating state because of nearly saturated hydrogenation (Supplementary Information Fig. S3 and S14), which agree well with the different H concentrations revealed by secondary-ion mass spectrometry measurement (Fig. S15).”

Figures: Depth profiles of H⁺ and V⁴⁺ ions from the initial pure VO₂ film and the hydrogenated VO₂ films, measured with secondary-ion mass spectrometry (SIMS).

(5) What are the migration pathways through the monoclinic lattice (work by Mazhos in Phys. Chem. Chem. Phys., 2017,19, 22538-22545 may provide some useful ideas on possible diffusion pathways).

----Thanks for this helpful comment. Yes, it is interesting to investigate the migration pathways of H atoms in VO₂ lattice. The mentioned reference [Phys. Chem. Chem. Phys., 2017,19, 22538-22545] systematically investigated thermodynamic, electronic, and kinetic properties associated with the insertion of Li, Mg and Al atoms into rutile VO₂. The intercalation of H atoms in VO₂ should have the similar behavior as Li if considering the small atom radius and the valance state. We have also found some previous literatures [Nano Lett., 2014, 14, 5445–5451; J. Am. Chem. Soc. 2014, 136, 8100–8109] which studied the H diffusion behavior in VO₂ lattice. It was revealed that the hydrogen atoms preferred to diffuse along the c axis of rutile (a-axis of monoclinic) VO₂, along the oxygen “channels”.

Meanwhile, using the nudged elastic band (CI-NEB) calculations for transition states, we have examined in theory the H migration pathway in VO₂. From the energy profiles of H diffusion along the [100] and [011] direction, we found the energy barrier values as 0.8 and 1.4 eV, respectively. This suggests a most likely H migration pathway along the [100] direction.

Figures| (a) The diffusion pathway from surface to subsurface is depicted in the inset graph. Gray, red, purple beads stand for V, O, H atoms, respectively. Focusing on the first diffusion step from surface to subsurface (inset graph), the diffusion energy barrier decreased with the increasing of doped electrons in VO₂. (b) The H migration pathways along [100] direction as found by NEB transition state simulations. (c) The energy profile of the diffusion pathway along the [100]

direction. (d) The energy profile of the diffusion pathway along the [011] direction

In the revised paper, please refer to **red texts in Line 3-5/Paragraph 1/Page 3 of the main text**: “*Previous studies examined the thermodynamic and kinetic properties of H or Li doping in VO₂ lattice, showing that H atoms preferred to diffuse along the c-axis of rutile VO₂ or a-axis of monoclinic VO₂ (Ref 27: Nano Lett., 2014, 14, 5445–5451; Ref 28: Phys. Chem. Chem. Phys., 2017,19, 22538-22545).*” **and red texts in Line 5-7/Paragraph 2/Page 6 of the main text**: “*The doped electrons also promote the diffusion of surface hydrogens into the VO₂ crystal, with a possible migration pathway along the [100] direction (Supplementary Information Fig. S10)*”.

(6) Can the authors provide a balanced equation for the hydrogenation reaction? I am having trouble seeing how charges and protons balance across the entire construct and the simplistic schematic in Figure 4 does not provide a clear answer.

---We're grateful to the reviewer for this very helpful advice which indeed improved our presentation.

Please refer to the **red texts in captions of Figure 4d in main text**: “*Once the metal/VO₂ is immersing into acid solution, chemical reactions go sequentially as (a) $M-e \rightarrow M^+$; (b) $VO_2+x[e]+x[H^+] \rightarrow H_xVO_2$. Here protons penetrate to meet electrons, creating conductive H-doped structure. Meanwhile the attached metal (M) is gradually dissolved in acid to become M^+ cations for balancing charges in solution*”

(7) Are the interstitial species H⁺ or H radicals reduced one electron?

Since the work is not really the first report of hydrogenation of VO₂, a more detailed mechanistic perspective rather than ascribing the observed behavior to a work function difference ought to be necessary.

----Yes, for the hydrogenation reaction, it is the protons (H⁺) in acid solution that penetrate into VO₂ lattice, become neutral H atoms if meets with electrons in lattice, and consequently form O–H species, which pronouncedly modulate the phase structure of VO₂ film.

We agree with the reviewer that this work is not the first report of hydrogenation of VO₂, and it is necessary to clarify the underlying mechanism. Here the use of work function difference is just to clarify the driving force of charge donation into VO₂, as demonstrated by our experimental observations of Al/Zn with very low work function cause heavily hydrogenated VO₂ with insulating state, Pt/Au with high workfunction induce no hydrogenation, and Ag/Cu with work function in between result in lightly hydrogenated VO₂ with metallic state.

After that, we have illustrated in Figure 4a how the charges accumulated in VO₂

lattice attract hydrogen atoms to penetrate the crystal, in Figure 4b how hydrogen doping with small concentration converts the typical semi-conductor density of state (DOS) of pristine VO₂ to the DOS of metallic system, and in Figure 4c how lightly hydrogenated H_{0.25}VO₂ donates charges to pristine VO₂. All these together were summarized in **Figure 4d**, which presents the electron-hydrogen co-doping mechanism **in Paragraph 3/Page 7 of the main text**: *“The contagious spreading of electron-proton co-doping process is summarized in Fig. 4d: (I) Metal with lower workfunction donates electrons to the interfaced VO₂ due to Fermi level difference, resulted in accumulated extra electrons in oxide layer; (II) Doped electrons attract surrounding protons in acid solution to penetrate into the oxide semiconductor, creating H-doped structure...”*

As for how hydrogen doping affects VO₂ electronic structure, we have discussed the details from both theoretical and experimental point of view in our previous work (Ref. 29: PRB 96, 125130 (2017)). That’s why in this manuscript we focused on the report of a strikingly simple approach to hydrogenate monoclinic vanadium dioxide (VO₂) in acid solution at ambient condition by placing a small piece of suitable metal on VO₂ surface, which demonstrates a conceptually new atomic doping technique for VO₂ materials.

Reviewer #2 (Remarks to the Author):

The manuscript, entitled to “Non-catalytic hydrogenation of VO₂ in acid solution” by Chen et al., reported simple approach for hydrogenation of VO₂ without commonly used catalyst. Utilizing acid solution as a rich source of protons, complete hydrogenation of wafer-size VO₂ could be achieved by placing a small piece of appropriate metal on the VO₂ surface without corrosion of VO₂ by acid solution. To explain this interesting phenomena, the authors proposed an electron-proton co-doping mechanism, which is verified by the first principles calculations.

Overall, **this article shows unique non-catalytic hydrogenation approach** by controlling work function and electron transfer to VO₂ thin films. The result on non-catalytic hydrogenation using acid solution have not been reported before, although non-catalytic approach using atomic hydrogen gas (Nano Lett., 14, 5445 (2014)) or ionic liquid by gating. The role of electron transfer was emphasized by utilizing metal with low work function on VO₂ films, which can be further applicable to the development of new approach for band-filling control of strongly correlated VO₂ films. Therefore, this report will attract the attention of other researcher in materials science community and will potentially extend the scope of physics into the efficient band-filling control of correlated phase using hydrogen dopants.

----We appreciate the reviewer’s positive comments, which encouraged us a lot. Hopefully, “this report will attract the attention of other researcher in materials science community and will potentially extend the scope of physics...”

I have several technical comments on this manuscript, which should be addressed before the publication in Nature Communication.

1) The authors argued that H intercalation occurred in the VO₂ lattice after metal-acid treatment by showing the “indirect” methods (e.g., lattice expansion by XRD, resistance change by R-T measurement, valence change by XPS and XANES). But, those methods do not directly detect hydrogen atoms. Therefore, it is necessary to “directly” quantify the amount of hydrogen incorporated in thin films after hydrogenation. I would suggest using previously reported techniques, such as dynamic SIMS or RBS (See Nature 546, 124 (2017), Nature Mater. 15, 1113 (016)), as direct evidence of H intercalation in metal-acid treated VO₂ film.

---- We are very grateful to these helpful advices. SIMS or RBS method can more “directly” measure the exact amount of hydrogen incorporated in thin VO₂ films after hydrogenation. We have thus tested the depth profiles of H⁺ and V⁴⁺ ions from the initial pure VO₂ film and the hydrogenated VO₂ films, measured with secondary-ion mass spectrometry (SIMS). The results were shown in the following curves. It was observed that the H atoms are really doped into the VO₂ film through the metal-acid treatment.

Figures: Depth profiles of H⁺ and V⁴⁺ ions from the initial pure VO₂ film and the hydrogenated VO₂ films, measured with secondary-ion mass spectrometry (SIMS).

Please refer to the **red texts of Line 7-9/Paragraph 3/Page 11 in the main text:** *“To obtain direct information about the hydrogen concentration of the metal-acid VO₂ film sample, the secondary-ion mass spectrometry (SIMS) measurements (Quad PHI6600) were conducted.”* The measured SIMS results are displayed in the **Supporting Information Figure S15:** *“It was observed that quite low H atoms concentration was tested for the pure VO₂ film. While for the metal-acid treated VO₂ films, the H atoms concentration was much higher. It was clear that the insulator H-VO₂ showed higher H atoms doping. The interface between the VO₂ film and substrate can be clearly observed according to the H⁺ and V⁴⁺ ions curves”.*

2) The authors reported that all experiment using metal-attached VO₂ film, which showed anti-corrosion property, were performed in 2%wt H₂SO₄ acid with same pH condition. I am wondering if the authors tried doing same experiment by changing pH of H₂SO₄ acid.

----Thanks again for the helpful advice. Yes, it is really an interesting experiment. We have tested the experiments by 10%wt or even 20%wt H₂SO₄ acid, which shows that the anti-corrosion property is still observed. We have also tested the experiments by dilute hydrochloric acid or oxalic acid, which show the similar results.

Please refer to the **red texts of the caption of Figure S2 (Supporting Information):** *“In addition, we have tested the experiments by 10%wt and up to 20%wt H₂SO₄ acid, which shows that the anti-corrosion property is still observed. We have also tested the experiments by dilute hydrochloric acid or oxalic acid, which show the similar results.”*

3) The authors argued that, after annealing Li-VO₂ film, the Li atoms was driven out from the VO₂ lattice and congregated on the top-most surface of the film as shown in Fig. S15. I suspected the formation of Li oxide on the surface, instead of congregation of Li atoms on the surfaces. How could the author rule out this effect on the surface?

----Yes, we agree with the reviewer that it should be the Li oxide on the surface, while not the pure Li atoms. In fact, from Figure S16 (the revised version), we can see that after heating treatment, the metallic Li-doped VO₂ goes back to the initial M-VO₂ insulator (S16a and S16c). The XPS in S16d clearly shows the variation of Li 1s peak. Considering the surface sensitive of XPS technique, the greatly enhanced Li peak intensity after the annealing treatment indicates the increased concentration of Li at the top surface. According to the peak position of Li 1s peak at about 55.3eV in S16d, the Li element here should be Li oxide (Li⁺), while not pure Li atoms (The related binding energy ~54.9eV). Please refer to the red texts in the **caption of supporting information Figure S16:** *“while this peak becomes much stronger after annealed, showing the Li atoms are driven out from the VO₂ lattice and congregated on the*

surface to form some Li-oxides”.

4) Fig. 2c showed that whole 2-inch $\text{VO}_2/\text{Al}_2\text{O}_3$ wafers become metallic uniformly by Cu-acid treatment. I am wondering if wafer-scale uniform insulator-metal transition also occurs in other metal-acid with Al or Zn. Furthermore, do you have any data on the wafer-scale resistance mapping of fully hydrogenated insulating $\text{VO}_2/\text{Al}_2\text{O}_3$ wafers formed by metal-acid treatment, which can be only shown in the manuscript on resistance mapping of pristine VO_2 films (Fig. 2b) and metallic hydrogenated VO_2 (Fig. 2c)?

---Yes, it is really the fact that in other metal-acid treatment with Al or Zn, the whole 2-inch $\text{VO}_2/\text{Al}_2\text{O}_3$ wafers become fully hydrogenated insulating wafers, see the following image.

Please refer to the **supporting information Figure S14**. It can be observed that though the fully hydrogenated wafer is insulating, the color is quite different from the initial M- VO_2 insulator. The fully hydrogenated wafer shows light grey color with enhanced transparency. The resistance mapping is showing as the above pictures.

Figure: (Left) The resistance mapping of the fully hydrogenated insulating wafer; (Middle) the picture of the fully hydrogenated insulating wafers and (Right) The pristine M- VO_2 wafer.

5) Are all the samples in Fig. 1e, f, g, which is characterized by XRD, XPS, XANES, hydrogenated using steel (Fe) tweezers? Figure caption is unclear. If this was hydrogenated using Fe tweezers, was samples fully hydrogenated? In terms of XRD, XPS, XANES, samples seem to become fully hydrogenated insulating phase, but the work function of Fe is similar to Cu and is higher than Al/Zn. Therefore, I should expect less effective hydrogenation process of Fe-acid treatment than Al/Zn-acid treatment, but the effectiveness of Fe-acid hydrogenation appears to be similar to that of Al/Zn-acid hydrogenation (Please compare Fig 1e, f, g and Fig. S3).

---We thank the reviewer for bringing this into our attention, which helps a lot to improve our presentations. All the samples in Fig.1e~g are hydrogenated samples by

contacted Fe in acid solution. In the revised paper, we have revised the related figure caption, to make it much clearer.

From our current experiment, it is observed that by using Fe granule or Fe tweezers, the VO₂ film can really be fully hydrogenated after quite long time (e.g. 10 hours or longer). While by using Cu granule, we can only observe the metallic hydrogenated state, while not the fully hydrogenated insulator state even after quite long time (> 20 hours). However, for Al/Zn-acid treatment, the VO₂ film can be quickly fully/heavily hydrogenated into the insulating state after even one or two hours.

From these experimental observations, it is suggested that the metal granule with different work function value really determines the final hydrogenated VO₂ state. Fe granule (work function: 4.5~4.6eV) and Cu granule (work function: 4.6~4.7eV) maybe just located at the critical region for the metal-acid treatment to drive the transition from metallic hydrogenated state to fully hydrogenated insulator state.

6) Small typo: the blue line in Fig. 1e, which was shifted more than the red line from the peak of pristine sample, should indicate metal-acid treated samples for 10 hours, not metal-acid treated samples for 1.5 hours.

---- Thanks for pointing out these mistake. We have now corrected it in the revised manuscript.

Reviewer #3 (Remarks to the Author):

Chen et al. **presented a new way of hydrogenating VO₂** by using acid solution. **The method is novel and unique.** By taking advantage of low work-function metal, large-scale hydrogenation of VO₂ can be achieved. Overall, the experimental **result is interesting.** However, there are a few concerns that need to be addressed before publication.

----We appreciate the reviewer for these positive evaluations on our work.

1. The paper emphasizes on the hydrogenation of VO₂ film, but also mentioned that using acid might etch the surface. This argument is supported using concentration data (Figure 1d) of dissolved V ions. It will be more intuitive and straightforward to show the thickness change as a function of immersion time of VO₂ dipping in acid by using atomic force microscope.

----We thank the reviewer for this very helpful advice. We totally agree that by using AFM, it will be more intuitive and straightforward to show the thickness change as the function of immersion time. Thus, we have conducted the thickness test by AFM for the samples. Please refer to **supporting information Figure S2a and red texts in Line 11-14/Paragraph 2/Page 3 in main text: “In addition, the Atomic Force**

Microscope (AFM) measurements show nearly zero thickness variation for metal-acid treated samples (Fig. S2a), which is consistent with the SEM cross-section image and confirms the anti-corrosion ability.” In contrast, without the metal (Fe) contact, the film thickness is gradually decreasing due to the corrosion of VO₂ in acid solution.

Figure: The VO₂ film thickness variations as the function of immersing time in acid solution with and without metal contact, which were tested by AFM method.

2. The authors claim that it may also work for other oxides. This assertion should be deleted unless they can demonstrate the same phenomenon in different oxides, such as ZnO, or TiO₂, etc. Comparing to other oxides VO₂ is unique because the strong correlation of electrons makes it a model system of Mott insulator. Many properties are rooted from this aspect.

---We thank the reviewer for pointing out this issue, which makes our statement more reasonable. We agree with the reviewer that the observation in the current experiment is closely associated with the special phase transition property of VO₂. Thus, in the revised paper, we have deleted the related words.

3. In line 78, the authors claim that the anticorrosion of VO₂ is due to the hydrogenation. So the question is that at the very beginning of the reaction, the metal is just in contact with the film and the hydrogenation hasn't started yet, is it still anticorrosive? What would be the dynamics of hydrogenation caused anticorrosion or how the anticorrosion ability evolves during the hydrogenation process?

---Thanks the reviewer for bringing this into our attention, as the dynamics of hydrogenation determines the working of our proposed mechanism. Please refer to red texts in **Line 1-4/Paragraph 1/Page 8 in main texts**: “It should be pointed out that since the corrosion of VO₂ caused by oxygen atom moving out of lattice is much slower than the migrations of electrons or protons, the dynamics of this co-doping mechanism ensure the quick hydrogenation of VO₂ surface before being corroded by

acid, resulting in the anti-corrosion property of wafer-size VO₂ film even at the first beginning stage” .

In fact, for bare VO₂ film, it reacts with acid solution and dissolve (or corrode) quickly, following the reaction route: VO₂ + 4H⁺ → V⁴⁺ + 2H₂O. While if a suitable metal (M) is in contact with the VO₂ film, charges will flow immediately from metal to VO₂. When the metal/VO₂ is immersed in acid solution, the reaction route will change completely: VO₂ + x[e⁻] + xH⁺ → H_xVO₂; M-x[e⁻] → M^{x+}. Under this condition, the movement of hydrogen atoms and electrons are much faster than the moving out of oxygen atoms in the lattice. Therefore, the top surface of VO₂ film becomes hydrogenated very quickly to be H_xVO₂ film with the changed phase structure, which obtained the “anticorrosive” ability immediately. This hydrogenation reaction starts on the top surface of VO₂ film (protect the whole film quickly), and then the H atoms diffuse to deeper layer so as to complete the thin VO₂ film hydrogenation. This situation can also be reflected from the following time-dependent Raman and XRD characterizations.

Figure: Raman (a) and XRD (b) characterizations along with the time-dependent meta-acid treatment.

4. In Figure 1e, the blue color line should correspond to 10 hours treatment, not the 1.5 treatment.

----Thanks for pointing out the mistake. We have now corrected it in the revised manuscript.

5. In line 99, the authors immersed metal together with the film, which means the metal also contacts with the acid directly. Here brings the question: will there be any possibility that metal dissolves into the acid and releases metal ions, which stops the

corrosion of VO₂? The authors need to rule out this possibility by performing experiment that half of the film contacted with metal left outside of the liquid and the other half from the same piece of film immerses into the acid to see if the hydrogenation can still be observed.

---We appreciate the reviewer for the helpful advice. It is the fact that the corrosion of VO₂ in acid solution follows the chemical reaction: $\text{VO}_2 + 4\text{H}^+ \rightarrow \text{V}^{4+} + 2\text{H}_2\text{O}$. While the “anticorrosive” property of VO₂ film is due to the formation of hydrogenated H_xVO₂ based on the reaction route: (a) $\text{M} - \text{e}^- \rightarrow \text{M}^+$; (b) $\text{VO}_2 + x[\text{e}^-] + x[\text{H}^+] \rightarrow \text{H}_x\text{VO}_2$ (Here “M” stands for metal), As we have described in the red texts of **Line 7-9/Paragraph 3/Page 4 of the main text**: “when the Cu particle is taken away after the treatment, the film remains stable in acid solution as it has already been hydrogenated (Supplementary Information Fig. S4).” Here please note that the hydrogenation of our metal-acid samples were demonstrated by XRD, XPS, XANES, SIMS, Raman characterizations. Moreover, the hydrogenated VO₂ after conventional precious metal catalysis also exhibit anticorrosive property.

According to the reviewer’s suggestion, we have conducted the experiment as shown below, which has also been added into the Supporting Information as Figure S17.

Figure: No anti-corrosion property if immersing only parts of a Cu/VO₂ system into acid without metal in solution.

This test shows that if we put half of the film contacted with Cu metal left outside of the liquid, the other half from the same piece of film in the acid will be dissolve (corroded) soon. It indicates that the balancing of charges in solution is essential for hydrogenating VO₂ in the metal-acid treatment.

Please refer to the **red texts in captions of Figure 4d in main text**: “Once the metal/VO₂ is immersing into acid solution, chemical reactions go sequentially as (a) $\text{M} - \text{e}^- \rightarrow \text{M}^+$; (b) $\text{VO}_2 + x[\text{e}^-] + x[\text{H}^+] \rightarrow \text{H}_x\text{VO}_2$. Here protons penetrate to meet electrons, creating conductive H-doped structure. Meanwhile the attached metal (M) is

gradually dissolved in acid to become M^+ cations for balancing charges in solution."

6. English need to be polished. For example, the sentence in line 114 is confusing. Single and uniform hydrogenated VO₂ film can be achieved by removing copper metal after metal-acid treatment. Line 59, "nature" needs to be changed to "natural". Line 67, "magically" needs to be deleted to fit into a writing style of scientific research paper.

---Thanks for pointing out this issue. We have gone through the whole manuscript and spent efforts to improve the English writing.

7. In line 170, the mechanism needs to be thought carefully. The authors propose a contagious spreading mechanism of hydrogenation. Does that mean the hydrogenation is spreading from the center of metal? Is there any proof of that? In contrast to this theory, a global hydrogenation process may happen at the film surface, because electrons transferred from metal to VO₂ surface and should expend to the whole surface instantaneously, not like protons which have much less mobility. Although monoclinic VO₂ is regarded as an insulator, it is still quite conductive comparing to conventional insulators, therefore, electrostatically charge can move easily on M-VO₂. Therefore, more experimental evidence are needed to approve the mechanism authors proposed.

---We thank the reviewer for bringing out this interesting comment, which drives us to think more clearly about the proposed mechanism.

We agree with the reviewer that the conductive surface of VO₂ film may deliver electrons to a large area of surface instantaneously. However, we also note that the migration of electron in semiconductor crystal is always limited by the mean free path (or diffuse length) of electron, which is normally in the scale of μm . Therefore, one can expect electrons donated by metals are firstly spreading up to limited distance away from the metal particle, which are then hydrogenated and become metallic to collect much more electrons. Starting from the newly hydrogenated area, electrons would be delivered further to enable more hydrogenation. The step-by-step spreading picture is reflected by the measured resistance distribution map of the metal (Cu)-acid treated VO₂ sample in Fig. 2c, where the onion-like contour map of resistance suggests a gradual spreading of hydrogenation treatment.

Following this helpful advice, we have also conducted Raman and XRD characterizations (Supporting information Figure S18) along with the time-dependent meta-acid treatment. It is observed that as the meta-acid treatment time increasing, the M-VO₂ film was actually converted gradually to the fully hydrogenated state. From the variation of Raman peak intensity as well as the shift of XRD, it is deduced that the hydrogenation process is gradually completed from the top layer to bottom parts.

Accordingly, we have revised the proposed mechanism with more details. Please refer to the **red sentences of Paragraph 3/Page 7 in main text**.

Figure: Raman and XRD characterizations along with the time-dependent meta-acid treatment.

8. In line 178 and 179, the authors conducted an experiment by continuously ramping the temperature during the hydrogenation, including the temperature above the phase transition temperature (67 degree). It complicates the experiment by mixing acid-hydrogenation and intrinsic phase transition because above 67 degree, the metallic phase of VO₂ will certainly cause dramatic hydrogenation due to the more opening rutile lattice structure as discovered by other researchers. So the authors should conduct another experiment by keeping the temperature at room temperature during the acid-hydrogenation, and take the sample out of acid for inspection of metal insulating transition by ramping the temperature. Repeated this process multiple times at each stage of hydrogenation.

---We are grateful to the reviewer for useful advices aiming at improving the quality of our work. It should be clarified that our experiments in Figure S14a~b are conducted exactly as the reviewer suggested.

In our experiment, we hydrogenated two VO₂ film samples by metal-acid treatment with Al and Zn at room temperature respectively. After quite long immersing time, the two samples were found to be fully hydrogenated into the insulating phases. Then we take them out of acid solution, and conduct the Resistance-Temperature measurement by ramping the temperature. The results are shown in Figure S14a~b. It can be observed that these H-doped insulating states were gradually converted to metallic H-doped phase and finally back to the original insulating M-VO₂ due to the hydrogen desorption from VO₂ crystal during the R-T testing cycles.

Thanks to the comment, we have now corrected the **caption of Figure S14** to make the description clearer.

Reviewers' comments:

Reviewer #1 (Remarks to the Author):

The revised manuscript makes a more compelling case for publication in Nature Communications-the primary argument now appears to be ease of hydrogenation of the VO₂ lattice rather than novelty in interstitial VO₂ doping. The addition of extensive first-principles calculations has furthermore provided a firmer physical basis to explain the phenomena observed here. I have a few remaining comments:

(a) The authors note in their letter: “the hydrogenated VO₂ film shows a rutile-like phase structure and the lattice parameters are sensitive to the H-doping concentration”. The first comment is most plausible in light of the Raman experiments that have now been included but what is the experimental evidence of the latter comment about H-doping concentration? Is it simply past XRD studies? Could the authors at least perform a Pawley fit for undoped and hydrogenated VO₂ and report how the lattice parameters are modified?

(b) I appreciate the addition of Table S3 that denotes several possible H-adsorption sites. However, H-incorporation presumably first occurs in the monoclinic and not the rutile phase. Can the authors perform a similar analysis for the monoclinic phase? I cannot clearly tell if that is the intent of Table S2-if it is I would ask that the authors sketch out the different H positions as they've done in Table S3.

(c) The addition of SIMS data greatly improves the quality of the manuscript. Is there any evidence for [V-OH]ⁿ⁺ or V³⁺? Why is the H⁺ concentration higher in insulating H-VO₂ as compared to metallic H-VO₂?

(d) The discussion of the O K-edge XANES spectra is somewhat imprecise. The observed changes in lineshape are not just a result of electron occupancy but also reflect loss of the d-parallel state upon metallization (see for example Nano Lett., 2013, 13 (10), pp 4857–4861)

Reviewer #2 (Remarks to the Author):

I have read the response letters from the authors on the referees' comments. The authors have addressed all my technical concerns. I recommend to accept this paper.

Responses to Reviewers:

Reviewer #1 (Remarks to the Author):

The revised manuscript makes a more compelling case for publication in Nature Communications-the primary argument now appears to be ease of hydrogenation of the VO₂ lattice rather than novelty in interstitial VO₂ doping. The addition of extensive first-principles calculations has furthermore provided a firmer physical basis to explain the phenomena observed here.

----We are very grateful to the reviewer for the encouraging comments.

I have a few remaining comments:

(a) The authors note in their letter: “the hydrogenated VO₂ film shows a rutile-like phase structure and the lattice parameters are sensitive to the H-doping concentration”. The first comment is most plausible in light of the Raman experiments that have now been included but what is the experimental evidence of the latter comment about H-doping concentration? Is it simply past XRD studies? Could the authors at least perform a Pawley fit for undoped and hydrogenated VO₂ and report how the lattice parameters are modified?

----We thank the reviewer for bringing out this issue. Yes, it is clear that from the shifts of XRD peak, we can observe the lattice expansion as the H doping concentration increasing, though there exist some intermediated phase structures.

In fact, from XPS results, we can estimate the H concentration in H_xVO₂ sample. In the following Figure-(a), the O 1s peak at 530.1 eV is from vanadium oxide, and the peak at 531.9 eV originates from the –OH species [*Nat. Nanotechnol.* 7, 357 (2012); *Nat. Mater.* 15, 1113 (2016); *Phys. Rev. B* 96, 125130 (2017)]. For hydrogenated VO₂, the –OH peak becomes stronger, indicating the formation of –OH bonds. Figure-(b) shows the changes of intensity ratio between the –OH peak and V-O peak before and after the hydrogenation. From these results, we can then estimate the hydrogen concentration in the samples. For an idea H_xVO₂ compound, the number of –OH bonds is x, while the V-O bonds should be (2-x). According to the above ratio values, we can estimate the current metallic phase sample has an estimated

stoichiometry of $H_{0.48}VO_2$, while for the insulating phase, we find $H_{0.89}VO_2$. Anyway, we note that it is difficult to determine the exact H-atom concentration for the metallic or insulating phases by XPS since the measurement is strongly dependent on the individual samples themselves and some intermediate phases may exist. However, these calculations from XPS results can be consistent with the secondary-ion mass spectrometry (SIMS) measurements for H atoms detection if just considering the H concentration ratio between the metallic and insulator phases.

Figure: (a) XPS peaks as well as the curves-fittings of O1s peaks for three different samples. O1s peaks can be fitted by two peaks related to O-V bonds and -OH bonds (b)The intensity ratio values of -OH peak and O-V peak. From this intensity ratio of XPS peak, the H atoms concentration can be estimated for the samples; (c)The XRD variations from pure VO₂ to metallic H-VO₂ and then to insulating H-VO₂ states. Some intermediated phases can be observed. (d)Based on the XRD and XPS results, it is possible to examine the crystal lattice variation as the function of H concentration. A linear relation can be drawn between the lattice volume and the x value in H_xVO₂

phase.

In addition, as reported in the recent literature [Ref. 4, Nat. Mater. 15, 1113-9 (2016)], for the VO₂(020)/Al₂O₃(0001) epitaxial film, the lattice of b-axis increased greatly (up to ~10%) during the hydrogenation process, where the a-axis and c-axis changed less by 1% due to the substrate effect. For a roughness estimation, the whole lattice volume variations can be taken on the major attribution of expanded axis b. Thus, based on the XRD variations in Figure-(c), we just plot the unit-cell volume as the function of x value in H_xVO₂ in Figure-(d). For comparison, the pure VO₂ and the fully hydrogenated HVO₂ prepared by traditional catalyst-assisted annealing method are also plotted. It can be observed that the volume of the crystal lattice of H_xVO₂ is approximate linear to the H concentration, which agrees with previous reports [Nat. Mater. 15, 1113-9 (2016)].

(b) I appreciate the addition of Table S3 that denotes several possible H-adsorption sites. However, H-incorporation presumably first occurs in the monoclinic and not the rutile phase. Can the authors perform a similar analysis for the monoclinic phase? I cannot clearly tell if that is the intent of Table S2-if it is I would ask that the authors sketch out the different H positions as they've done in Table S3.

----We thank the reviewer for bringing out this issue. Actually, this work mainly focuses on the monoclinic phase of VO₂ in both experiments and simulations, which is why we have only presented the simulation results of the monoclinic unit cell in Table S2 and S3. Since the atomic positions of Oxygen in the monoclinic unit cell are not equal, there are several possible interstitial sites for H atom to reside in the V₄O₈ cell, as denoted in Table S3 of the Supporting information. It is of course interesting to check the rutile phase. For rutile phase, all O atoms in a unit are equal because of high symmetry, so there is only one interstitial site for H atom in the rutile V₂O₄ unit. And we have done calculations to check the binding energy of H intercalation. Please refer to the atomic models illustrated below the Table S3 of the Supporting information.

(c) The addition of SIMS data greatly improves the quality of the manuscript. Is there any evidence for $[\text{V-OH}]^{n+}$ or V^{3+} ? Why is the H^+ concentration higher in insulating H-VO_2 as compared to metallic H-VO_2 ?

----Yes, we thank the reviewer's positive comment about the SIMS results. For SIMS testing, it is really difficult to clarify the valence state such as $\text{V}^{3+}/\text{V}^{4+}$ during the experiment. The chemical state of V atoms can be clearly detected by XPS as shown in Figure 1f. The $[\text{V-OH}]^{n+}$ clusters are also hard to be detected due to the high energy of Cs^+ primary ions.

The H^+ concentration is higher in insulating H-VO_2 as compared to metallic H-VO_2 due to the different hydrogenation degree. Our results indicate that lower H-doping concentration in VO_2 results in the metallic state phase, while much higher H concentration will lead to an insulating phase structure as showing in the following image:

(d) The discussion of the O K-edge XANES spectra is somewhat imprecise. The observed changes in lineshape are not just a result of electron occupancy but also reflect loss of the d-parallel state upon metallization (see for example Nano Lett., 2013, 13 (10), pp 4857–4861)

----yes, we thank the reviewer for pointing out this issue. We agree with the reviewer's comment that the lineshape changes in O K-edge XANES spectra also reflect the loss of the d-parallel state upon metallization. In the revised paper, we have added this point as well as the above reference (See the Ref.30 in the main text).

Reviewer #2 (Remarks to the Author):

I have read the response letters from the authors on the referees' comments. The authors have addressed all my technical concerns. I recommend to accept this paper.

---We are very grateful to the reviewer's recommendation.

Reviewer #1 (Remarks to the Author):

I am satisfied with the revisions and am happy to recommend publication in Nature Communications.

Responses to Reviewers:

REVIEWERS' COMMENTS:

Reviewer #1 (Remarks to the Author):

I am satisfied with the revisions and am happy to recommend publication in Nature Communications.

---We thank referee #1 for recommendation of publication of our manuscript.

List of changes

1) The abstract of the paper has been shortened no more than 150 words and does not contain references.

2) Various minor revisions have been made according to editor's suggestions.